# FSH-blocking therapeutic for osteoporosis

**Sakshi Gera**[1†], **Tan-Chun Kuo**[1†], **Anisa Azatovna Gumerova**[1†], **Funda Korkmaz**[1†], **Damini Sant**[1], **Victoria DeMambro**[2], **Karthyayani Sudha**[1], **Ashley Padilla**[1], **Geoffrey Prevot**[3], **Jazz Munitz**[3], **Abraham Teunissen**[3], **Mandy MT van Leent**[3], **Tomas GJM Post**[3], **Jessica C Fernandes**[3], **Jessica Netto**[1], **Farhath Sultana**[1], **Eleanor Shelly**[1], **Satish Rojekar**[1], **Pushkar Kumar**[1], **Liam Cullen**[1], **Jiya Chatterjee**[1], **Anusha Pallapati**[1], **Sari Miyashita**[1], **Hasni Kannangara**[1], **Megha Bhongade**[1], **Puja Sengupta**[1], **Kseniia Ievleva**[1], **Valeriia Muradova**[1], **Rogerio Batista**[1], **Cemre Robinson**[1], **Anne Macdonald**[1], **Susan Hutchison**[1], **Mansi Saxena**[4], **Marcia Meseck**[1,4], **John Caminis**[1], **Jameel Iqbal**[1], **Maria I New**[1], **Vitaly Ryu**[1], **Se-Min Kim**[1], **Jay J Cao**[5], **Neeha Zaidi**[6], **Zahi A Fayad**[3], **Daria Lizneva**[1], **Clifford J Rosen**[2], **Tony Yuen**[1*], **Mone Zaidi**[1*]

[1]Center for Translational Medicine and Pharmacology and The Mount Sinai Bone Program, Departments of Medicine and of Pharmacological Sciences, Icahn School of Medicine at Mount Sinai, New York, United States; [2]Maine Medical Center Research Institute, Scarborough, United States; [3]BioMedical Engineering and Imaging Institute, Icahn School of Medicine at Mount Sinai, New York, United States; [4]Tisch Cancer Institute, Icahn School of Medicine at Mount Sinai, New York, United States; [5]United States Department of Agriculture, Grand Forks Human Nutrition Research Center, Grand Forks, United States; [6]Sidney Kimmel Comprehensive Cancer Center, Johns Hopkins University, Baltimore, United States

**\*For correspondence:**
tony.yuen@mountsinai.org (TY);
mone.zaidi@mountsinai.org (MZ)

[†]These authors contributed equally to this work

**Abstract** Pharmacological and genetic studies over the past decade have established the follicle-stimulating hormone (FSH) as an actionable target for diseases affecting millions, namely osteoporosis, obesity, and Alzheimer's disease. Blocking FSH action prevents bone loss, fat gain, and neurodegeneration in mice. We recently developed a first-in-class, humanized, epitope-specific FSH-blocking antibody, MS-Hu6, with a $K_D$ of 7.52 nM. Using a Good Laboratory Practice (GLP)-compliant platform, we now report the efficacy of MS-Hu6 in preventing and treating osteoporosis in mice and parameters of acute safety in monkeys. Biodistribution studies using [89]Zr-labeled, biotinylated or unconjugated MS-Hu6 in mice and monkeys showed localization to bone and bone marrow. The MS-Hu6 displayed a β phase $t_{1/2}$ of 7.5 days (180 hr) in humanized Tg32 mice. We tested 217 variations of excipients using the protein thermal shift assay to generate a final formulation that rendered MS-Hu6 stable in solution upon freeze-thaw and at different temperatures, with minimal aggregation, and without self-, cross-, or hydrophobic interactions or appreciable binding to relevant human antigens. The MS-Hu6 showed the same level of "humanness" as human IgG1 in silico and was non-immunogenic in ELISpot assays for IL-2 and IFN-γ in human peripheral blood mononuclear cell cultures. We conclude that MS-Hu6 is efficacious, durable, and manufacturable, and is therefore poised for future human testing.

## Editor's evaluation

In this manuscript, the authors describe a comprehensive characterization of a new humanized FSH blocking antibody (MS-Hu6), which they have studied in depth in terms of its efficacy on bone. They

provide compelling data on mouse and monkey species with a complete evaluation of their pharmacokinetics and biodistribution and characterize their effect on the treatment of obesity and bone loss.

## Introduction

While osteoporosis is a disease of public health concern, the paucity of therapies to prevent and treat bone loss continues to represent a challenge (*International Osteoporosis foundation, 2022*; *World Health Organization, 2022*; *Genentech, 1998*). Particularly during the late perimenopause, precipitous bone loss accompanies the onset of visceral obesity, dysregulated energy balance, and reduced physical activity (*Sowers et al., 2006a*; *Sowers et al., 2001*; *Sowers et al., 2007*; *Sowers et al., 1996*; *Sowers et al., 2003a*; *Sowers et al., 2003b*; *Sowers et al., 2006b*). These aberrant physiological changes across the menopausal transition are not fully explained by low estrogen, as estrogen levels are relatively unperturbed, while serum follicle-stimulating hormone (FSH) levels rise to maintain estrogen secretion from an otherwise failing ovary (*Randolph et al., 2004*; *Randolph et al., 2011*; *Randolph et al., 2003*). Furthermore, serum FSH, bone turnover, and bone mineral density (BMD) correlate well during late perimenopause [review: (*Zaidi et al., 2018*)]. Activating *FSHR* polymorphisms in post-menopausal women are linked to a high bone turnover and reduced BMD (*Rendina et al., 2010*). The question has been whether a rising FSH level drives peri-menopausal and perhaps even post-menopausal bone loss. In 2006, we provided the first evidence for a direct action of FSH on the bone (*Sun et al., 2006*). Since then, despite controversy fueled mainly by the overinterpretation of clinical studies with GnRH agonists that suppress not only FSH but also GnRH and LH (*Drake et al., 2010*), there is replicable evidence that the selective inhibition of FSH action in mice, for example by using novel FSH-blocking antibodies or a glutathione S-transferase-FSH fusion protein as a vaccine, protects against hypogonadal bone loss (*Ji et al., 2018*; *Liu et al., 2017*; *Zhu et al., 2012a*; *Zhu et al., 2012b*; *Geng et al., 2013*). Thus, it makes both biological and clinical sense to selectively inhibit FSH action to prevent bone loss.

We and our collaborators have also shown that inhibiting FSH by FSH-blocking antibodies reduces white adipose tissue in every fat compartment, induces thermogenic (or beige) adipose tissue, and increases energy expenditure in mice (*Liu et al., 2017*). Reduced fat mass has also been documented with a vaccine containing tandem repeats of the 13-amino-acid-long FSH receptor-binding FSHβ sequence to which our antibodies were raised (*Han et al., 2020*). An interventional study in treatment-naïve prostate cancer patients comparing orchiectomy versus triptorelin showed that, with near-zero testosterone, patients on triptorelin (reduced serum FSH and LH) had significantly lower body weight and fat mass compared to those post-orchiectomy (*Østergren et al., 2019*). Even recognizing the constraints of using a GnRH agonist, this dataset suggests that lowering serum FSH could, in principle, have beneficial effects on body composition in people, despite concomitant reductions in GnRH and LH. There is also new evidence that selective FSH blockade lowers serum total and LDL cholesterol and prevents neurodegeneration in mice (*Guo et al., 2019*; *Song et al., 2016*).

We hypothesize that blocking FSH action will reduce bone loss in people and may have additional benefit in preventing body fat accumulation, hyperlipidemia, and neurodegeneration. Toward this goal, we have developed our lead candidate, a first-in-class humanized FSH-blocking antibody, MS-Hu6. The latter binds a 13-amino-acid-long epitope of human FSHβ (LVYKDPARPKIQK) with high affinity, and by doing so, blocks the interaction of FSH with its receptor (*Gera et al., 2020*). Here, we report a comprehensive characterization of MS-Hu6 in terms of its in vivo prevention and treatment efficacy in mouse models of osteoporosis, acute safety in monkeys, full evaluation of pharmacokinetics and biodistribution, and a compendium of its physicochemical properties. This new information provides the framework for first-in-human studies toward the future use of MS-Hu6 in osteoporosis.

# Results

## Anabolic efficacy of MS-Hu6 in preventing and treating osteoporosis in mice

In choosing MS-Hu6 as the lead candidate from an array of 30 humanized clones, we examined the electrostatic binding in silico and determined $K_D$ by surface plasmon resonance in vitro (*Gera et al., 2020*). The MS-Hu6 had the best affinity ($K_D$ = 7.52 nM), approaching that of trastuzumab. We fine-mapped the three top candidates to document subtle differences in binding modes (*Gera et al., 2020*). In addition, we established that MS-Hu6 blocked the binding of labeled recombinant human FSH to the FSHR, and in doing so, inhibited osteoclastogenesis (*Gera et al., 2020*).

Male mice were injected with MS-Hu6 or human IgG (7 µg/day, 5 days a week) for 8 weeks. The latter dose was based on the in vitro $IC_{50}$ of MS-Hu6 in inhibiting osteoclastogenesis in mice, which was ~30-fold lower than our polyclonal antibody (*Gera et al., 2020*). Histomorphometry of femoral metaphysis and spine (L1-L3) showed significant increases in fractional bone volume (B.Ar/T.Ar) and trabecular thickness (Tb.Th), without an effect on trabecular number (Tb.N) (*Figure 1A and C*, *Figure 1—source data 1*). Dynamic histomorphometry of both the femoral metaphysis and spine showed evidence for increased mineral apposition rate (MAR) and bone formation rate (BFR) (*Figure 1B and D*, *Figure 1—source data 1*), confirming increased bone formation at both sites. Osteoclast surface (Oc.S/BS) was not reduced at this dose (*Figure 1B and D*, *Figure 1—source data 1*). The latter finding is not surprising as male mice were not in a high bone turnover state, in which instance a further lowering of bone resorption from baseline would not normally be expected. Of note, after 8 weeks of MS-Hu6, there was a drop (p=0.058) in serum activin, but serum FSH, LH, and inhibin were not significantly altered (*Figure 1E*, *Figure 1—source data 1*). The absence of a reduction in serum FSH is not unexpected, as MS-Hu6 only blocks the interaction between FSH and the FSHR, and serum FSH levels have remained unchanged in previous studies with our polyclonal FSH-blocking antibody (*Zhu et al., 2012a*). In all, the data clearly document an anabolic action of MS-Hu6 in mice.

To further explore the anabolic action of MS-Hu6, and to replicate our dataset in C.J.R.'s lab, 6-week-old female mice were ovariectomized, followed at 24 weeks of age by injection of MS-Hu6 or human IgG, daily, at 100 µg/day, for 4 weeks and then 50 µg/day for a further 4 weeks. Total body and femoral BMD measured by *PIXImus* was increased significantly in mice treated with MS-Hu6 (*Figure 1F*, *Figure 1—source data 1*). The µCT of the femoral epiphysis showed increased BV/TV (p=0.079), Tb.N (p=0.057) and connectivity density (Conn.D; p<0.05), reduced trabecular spacing (Tb. Sp; p<0.05), and no change in Tb.Th (*Figure 1G*, *Figure 1—source data 1*). Cortical thickness (Ct. Th) also increased significantly with MS-Hu6 (*Figure 1G*, *Figure 1—source data 1*). Expectedly, these anabolic effects were not seen in similarly treated C3H/HeJ mice, which are known to display a high bone mass phenotype (*Figure 1—figure supplement 1*, *Figure 1—figure supplement 1—source data 1*; *Bouxsein et al., 2005*). Overall, therefore, our data show that consistent with previous studies using our polyclonal antibody (*Zhu et al., 2012a*), MS-Hu6 displays a potent action in stimulating new bone formation and thus replenishing lost bone. A further dose-finding studies are currently underway to determine the minimum effective and maximum tolerated doses of MS-Hu6 in relation to its anabolic action on bone.

## Pharmacokinetics of MS-Hu6

Pharmacokinetic studies were performed in three mouse models—C57BL/6, CD1, and Tg32 mice—using $^{89}$Zr-labeled, unconjugated, or biotinylated MS-Hu6. For $^{89}$Zr labeling, MS-Hu6 was incubated with the chelator DFO-p-NCS for 3 hr, followed by incubation with $^{89}$Zr-oxalate for 1 hr at 37°C, ultra-filtration (10 kDa cut-off), and thin layer chromatography for quality check (*van de Watering et al., 2014*; see Methods). $^{89}$Zr-MS-Hu6 was injected as a single dose of 250 µCi (~250 µg) into the retro-orbital sinus of 3-month-old male C57BL/6 mice (two combined experiments, N=5 mice each). For γ-counting few drops of blood were drawn from the tail vein at 5, 30, and 60 min, and then at 2, 4, 24, 48, 72, and 120 hr. There was an increase in serum $^{89}$Zr-MS-Hu6 levels to a $C_{max}$ of 17.4% injected dose per gram of blood, or ~87 µg/mL, which was followed by a gradual decay of radioactivity with a β phase $t_{\frac{1}{2}}$ of 29 hr (*Figure 2A*, *Figure 2—source data 1*).

As C57BL/6 mice are inbred strains, we attempted to validate the pharmacokinetic studies in an outbred strain—CD1. The latter mice display genetic diversity reminiscent of the human population and are used widely for toxicology and efficacy testing (*Annas et al., 2013*). For biotinylation,

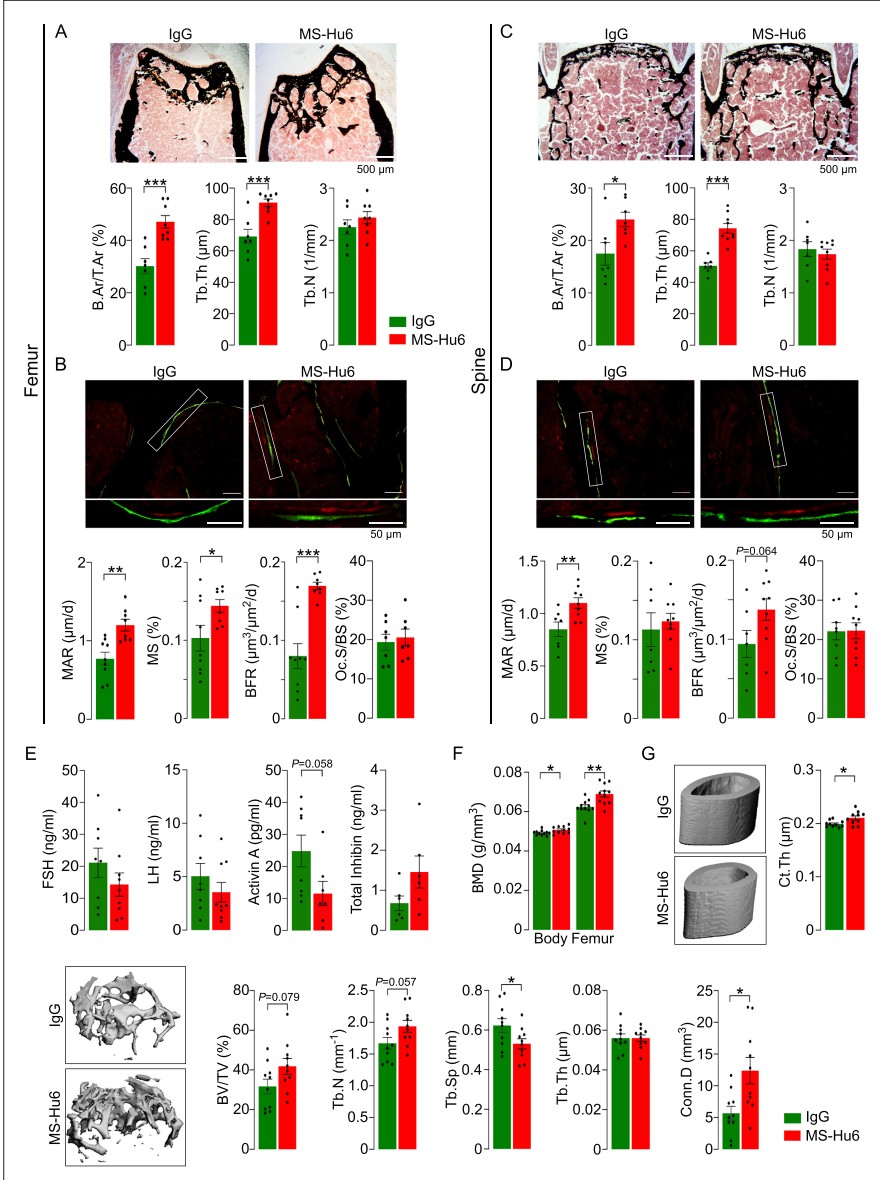

**Figure 1.** MS-Hu6 stimulates new bone formation and increases bone mass. Representative images of von Kossa-stained femoral epiphyses (**A**) and spine (**C**) of male FVB (Thermo) mice treated with MS-Hu6 or human IgG. Two-dimensional histomorphometric parameters showing bone volume (B.Ar/T.Ar), trabecular thickness (Tb.Th), and trabecular number (Tb.N). Dynamic histomorphometry showing representative images of double-labeled sections and quantitative data on mineral apposition rate (MAR), mineralizing surface (MS), and bone formation rate (BFR) at femoral epiphyses (**B**) and spine (**D**); osteoclast surfaces (Oc.S) are also shown. Shown also are serum follicle-stimulating hormone (FSH), luteinizing hormine (LH), activin A, and inhibin levels (**E**). Parallel studies carried out at C.J.R.'s lab used ovariectomized C57BL/6 mice, which were injected 18 weeks post-ovariectomy with MS-Hu6 or human IgG for 8 weeks. Shown are *PIXImus* measurements of total body and femur bone mineral density (BMD) (**F**), as well as µCT images and quantitative estimates of fractional bone volume (BV/TV), Tb.N, trabecular spacing (Tb.S) and Tb.Th, connectivity density (Conn.D), and cortical thickness (Ct.Th) (**G**) (performed at J.J.C.'s lab). Statistics: mean ± SEM; N=7, 8 mice/group for panel A; N=7–9 mice/group for panels B, C, and D; N=6–9 mice/group for panel E; N=11 mice/group for panel F; and N=10 mice/group for panel G; two-tailed Student's *t*-test (IgG versus MS-Hu6), *p<0.05, **p<0.01, ***p<0.001, or as shown.

The online version of this article includes the following source data and figure supplement(s) for figure 1:

**Source data 1.** Source data for *Figure 1*.

**Figure supplement 1.** Bone phenotype of the C3H/HeJ mouse is resistant to anabolic actions of MS-Hu6.

**Figure supplement 1—source data 1.** Source data for *Figure 1—figure supplement 1*.

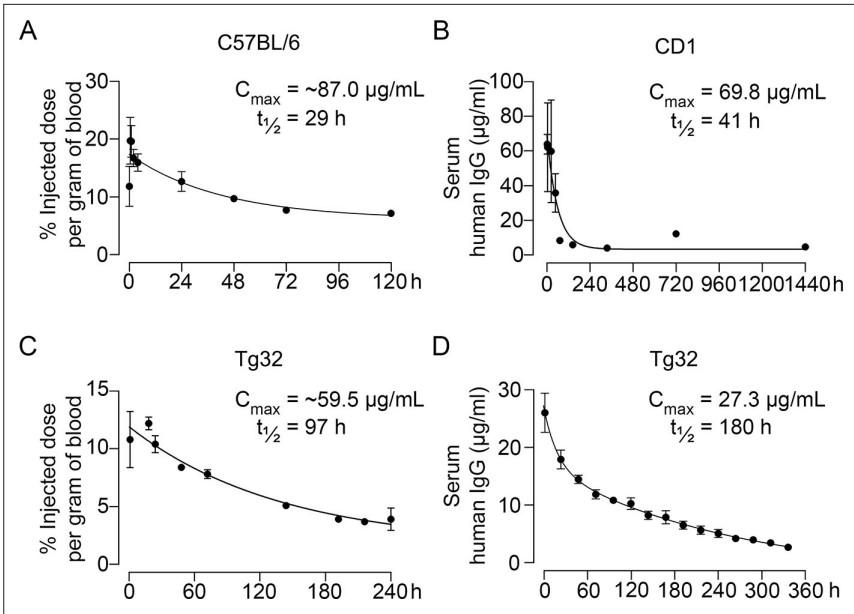

**Figure 2.** Pharmacokinetics of MS-Hu6. Plasma levels, $C_{max}$ and β phase $t_{1/2}$ values for MS-Hu6 injected into C57BL/6 mice (250 µCi of $^{89}$Zr-MS-Hu6, N=5 mice per experiment followed longitudinally, two combined experiments) (**A**), CD1 mice (200 µg biotinylated MS-Hu6, N=3–4 mice per time point) (**B**), Tg32 mice (250 µCi of $^{89}$Zr-MS-Hu6, N=5 mice followed longitudinally) (**C**), and Tg32 mice (200 µg unconjugated MS-Hu6, N=5 mice followed longitudinally) (**D**). Data are presented as mean ± SEM.

The online version of this article includes the following source data for figure 2:

**Source data 1.** Source data for *Figure 2*.

MS-Hu6 was incubated in the presence of N-hydroxysuccinimide ester-biotin in NaHCO$_3$ (pH 8), and the product was purified through ultrafiltration (10 kDa cut-off; see Methods). Biotinylated MS-Hu6 (200 µg) was injected intraperitoneally with blood sampling by cardiac puncture at 2, 4, 24, 48, 72, 144, 336, 720, and 1440 hr (N=3–4 mice per time point), and ELISA-based measurements through capture by streptavidin-HRP. This yielded a $C_{max}$ of 69.8 µg/mL and a β phase $t_{1/2}$ of 41 hr (*Figure 2B*, *Figure 2—source data 1*).

To enable mouse-to-human comparisons, we studied the pharmacokinetics of MS-Hu6 in Tg32 mice. These mice express the *FCGRT* transgene encoding the human FcRn receptor on chromosome 2 on an *Fcgrt$^{-/-}$* background. The Tg32 mice show decreased plasma clearance of *human* IgG-based therapeutics—thus, more closely mimicking human pharmacokinetics than C57BL/6 mice. We injected 3-month-old male Tg32 mice with $^{89}$Zr-MS-Hu6 (~250 µCi or 250 µg) into the retro-orbital sinus followed by sampling at 1, 18, 24, 48, 72, 144, 192, 216, and 240 hr (N=5 mice). Expectedly, the β phase $t_{1/2}$ increased to 4 days (97 hr), with a $C_{max}$ of 11.9% injected dose per gram of blood, or ~59.5 µg/mL (*Figure 2C*, *Figure 2—source data 1*). We further studied the profile of i.p. administered unconjugated MS-Hu6 in Tg32 mice by injecting a single bolus dose of 200 µg and measuring human IgG by an in-house sandwich ELISA in which anti-human Fc and Fab were used to capture and detect bound MS-Hu6, respectively (N=5 mice). This yielded $C_{max}$ of 27.3 µg/mL, with β phase $t_{1/2}$ of 7.5 days (180 hr; *Figure 2D*, *Figure 2—source data 1*). Of note is that the β phase $t_{1/2}$ for i.v. trastuzumab, currently in human use, is ~8.5 days in Tg32 mice (*Low et al., 2020*)—this latter β decay translates into 21-day dosing intervals.

## Biodistribution and excretion of MS-Hu6

To study the biodistribution of $^{89}$Zr-MS-Hu6, we performed PET-CT scanning of the above-treated mice at 24, 48, and 72 hr (N=5 mice per time point). The images show evidence of both decay of circulating radioactivity after 24 hr, and its persistence in multiple organs up to 72 hr (*Figure 3A*). Maximal retention in terms of standardized uptake values (SUVs, normalized to muscle) was noted in the liver, with persistence in regions of interest (ROIs), namely bone marrow, subcutaneous, and visceral white

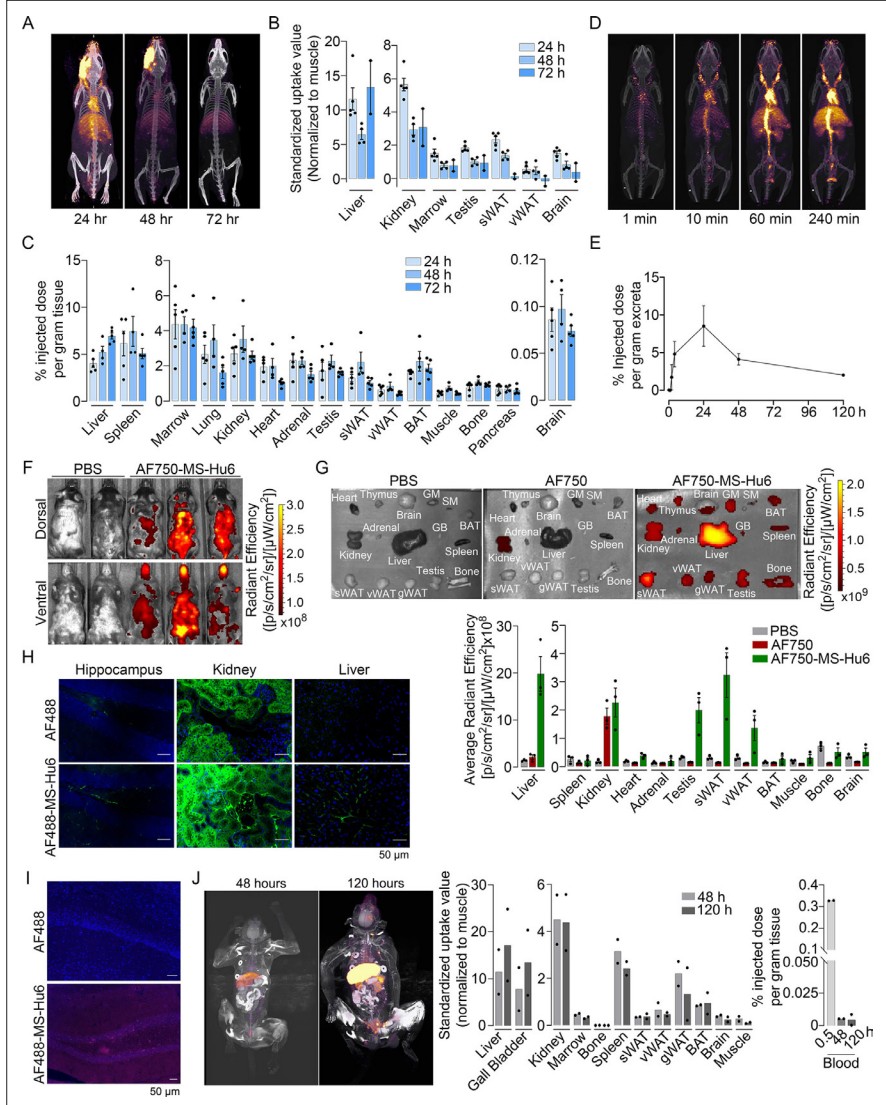

**Figure 3.** Biodistribution and excretion of MS-Hu6 in mice and monkeys. Representative PET-CT images of mice treated with a single bolus dose of $^{89}$Zr-labeled MS-Hu6 (250 µCi) at 24, 48, and 72 hours (**A**), together with quantitation in terms of standardized value uptake units (SUVs, normalized to muscle) in different organs (N=5, 4 and 2 mice for the three time points, respectively) (**B**). $^{89}$Zr-MS-Hu6 (γ-counts) in individual tissues isolated following perfusion of the mice with 20 mL PBS (N=5, 4 and 5 mice for the three time points, respectively) (**C**). Dynamic PET/CT images showing the uptake of $^{89}$Zr-MS-Hu6 over 240 min (**D**). Time course of excretion of $^{89}$Zr-labeled MS-Hu6 in feces (N=5 mice/time point) (**E**). Emitted whole body radiance on IVIS imaging of C57BL/6 mice injected with AF750-MS-Hu6 (200 µg) or PBS (**F**). IVIS imaging and quantitation (average radiance) of isolated perfused tissues, as shown, following AF750-MS-Hu6, AF750, or PBS injection (N=3 mice/group) (**G**). Immunofluorescence micrographs of hippocampal, kidney, and liver sections from C57BL/6 mice injected with AF488-MS-Hu6 or unconjugated AF488 (200 µg/mouse, i.v.) (AF488 - green; DAPI - blue) (**H**). Counterstaining with an anti-human IgG confirmed localization (AF488 - red; DAPI - blue) (**I**). Whole body PET/CT image, quantitation (SUVs) of multiple organs and serum radioactivity (γ-counts) following a single i.v. injection of $^{89}$Zr-MS-Hu6 (1.3 mg, ~1.3 mCi) in two *Cynomolgus* monkeys aged 14 and 15 years, respectively (**J**). Data are presented as mean ± SEM .

The online version of this article includes the following source data for figure 3:

**Source data 1.** Source data for *Figure 3*.

adipose tissue (WAT) depots, and the brain region (*Figure 3B*, *Figure 3—source data 1*). Of note is that these values reflect the presence of $^{89}$Zr-MS-Hu6 in both blood and tissues.

To determine the extent to which of $^{89}$Zr-MS-Hu6 persisted in individual tissues, we perfused the mice with 20 mL PBS before sacrifice and tissue isolation for γ-counting. Significant concentrations of $^{89}$Zr-MS-Hu6 were detected in multiple organs, including bone, bone marrow, subcutaneous WAT, visceral WAT, and brown adipose tissue (BAT) (*Figure 3C*, *Figure 3—source data 1*). Minimal amounts of $^{89}$Zr-MS-Hu6 were detected in isolated brain tissue at 72 hr—this is consistent with the low penetration of IgGs into the brain (0.05–0.1%; *Figure 3C*, *Figure 3—source data 1*). To hone into the early events, we monitored the uptake of $^{89}$Zr-MS-Hu6 by dynamic PET-CT imaging over 240 min. At 10 min, radioactivity was detected mainly in large vessels, which was followed at 60 and 240 min by permeation into organs (*Figure 3D*). As would be expected, radioactivity was not detected in the urine, but instead appeared in the feces (*Figure 3E*, *Figure 3—source data 1*).

To complement the $^{89}$Zr-based biodistribution studies, we labeled MS-Hu6 with Alexa-Fluor-750 (AF750), and injected C57BL/6 mice (N=3 mice) intravenously through the tail vein with AF750-MS-Hu6 (200 µg), AF750 alone or PBS. At 16 hr post-injection, anesthetized mice were imaged using the IVIS platform. We found significant soft tissue distribution of AF750-MS-Hu6 (*Figure 3F*). The mice were then perfused with PBS, followed by IVIS imaging of isolated tissue. Consistent with the $^{89}$Zr-based studies, there was uptake of AF750-MS-Hu6 by the liver, kidney, fat depots, bone, and brain (*Figure 3G*, *Figure 3—source data 1*). In contrast, in the AF750 (dye only) control group, localization was noted only in the kidney due to dye excretion, and not in other organs—in all, confirming organ retention of the AF750-MS-Hu6, and excretion of the unconjugated dye. Because of the expected minimal localization of MS-Hu6 in the brain, we further performed confirmatory studies by immunofluorescence. For this, we injected AF488-MS-Hu6 or unconjugated AF488 into the tail veins of C57BL/6 mice (200 µg per mouse). We detected immunofluorescence in the liver, kidney, and hippocampal sections in mice treated with AF488-MS-Hu6 (*Figure 3H*). In contrast, AF488-treated mice showed fluorescence in kidney sections, but not in the liver or brain (*Figure 3H*). Staining of hippocampal sections with an anti-human IgG confirmed localization (*Figure 3I*).

To understand MS-Hu6 biodistribution in a primate species, we injected $^{89}$Zr-MS-Hu6 as a single bolus dose (1.3 mg, ~1.3 mCi) into the tail veins of two male *Cynomolgus* monkeys aged 14 and 15 years, respectively. Blood was drawn via the tail vein at 0.5, 48 and 120 hr. $^{89}$Zr-MS-Hu6 peaked in the blood at 0.5 hr, with an expected decline, albeit with persistence in the serum, at 48 and 120 hr (*Figure 3J*, *Figure 3—source data 1*). PET/CT scanning revealed high SUV values in the liver and gall bladder, with lower SUVs in the kidney, spleen, fat depots, bone marrow, and the brain area (*Figure 3J*, *Figure 3—source data 1*).

## Acute and chronic tolerability of FSH blockade

We monitored standard safety parameters in treated monkeys for up to 100 min and did not observe significant acute or delayed changes post-injection in heart rate, respiratory rate, mean arterial blood pressure, systolic or diastolic blood pressure, or rectal temperature (*Figure 4A*, *Figure 4—source data 1*). We also drew blood on day 0 (pre-injection) and on days 2 and 5 post-injection. No concerning deviations from normative values were noted (*Figure 4B*, *Figure 4—source data 1*). This suggests that, albeit at a low dose, MS-Hu6 as a single intravenous bolus injection into monkeys appeared to be generally safe.

MS-Hu6 was generated by swapping the mouse framework region of our parent mouse monoclonal antibody Hf2 with the human IgG1 framework, keeping the CDR itself unaltered. While both Fc and Fab regions are human, mutations were introduced in the framework flanking the CDR region. Because MS-Hu6 is not fully "human," we first determined its "humanness" in silico by inputting the $V_L$ and $V_H$ sequences into abYsis. Comparison of Z-scores revealed a right shift in comparison with the human-mouse chimeric antibody (*Gera et al., 2020*), and in correspondence with human IgG1 (*Figure 4C*). In addition, we inputted the primary amino acid sequences of commercially utilized humanized ("zumab") and fully human ("mab") to find that the Z-scores fell in a narrow range away from our chimera or mouse IgGs (*Figure 4C*, Table). We next tested immunogenicity experimentally using ELISpot. The production of inflammatory cytokines IL-2 and IFN-γ in human peripheral blood mononuclear cell cultures was unaltered by MS-Hu6, in comparison with a standard CEFT peptide pool (positive control, Immunospot; *Figure 4D*, *Figure 4—source data 1*). Finally, we provide genetic

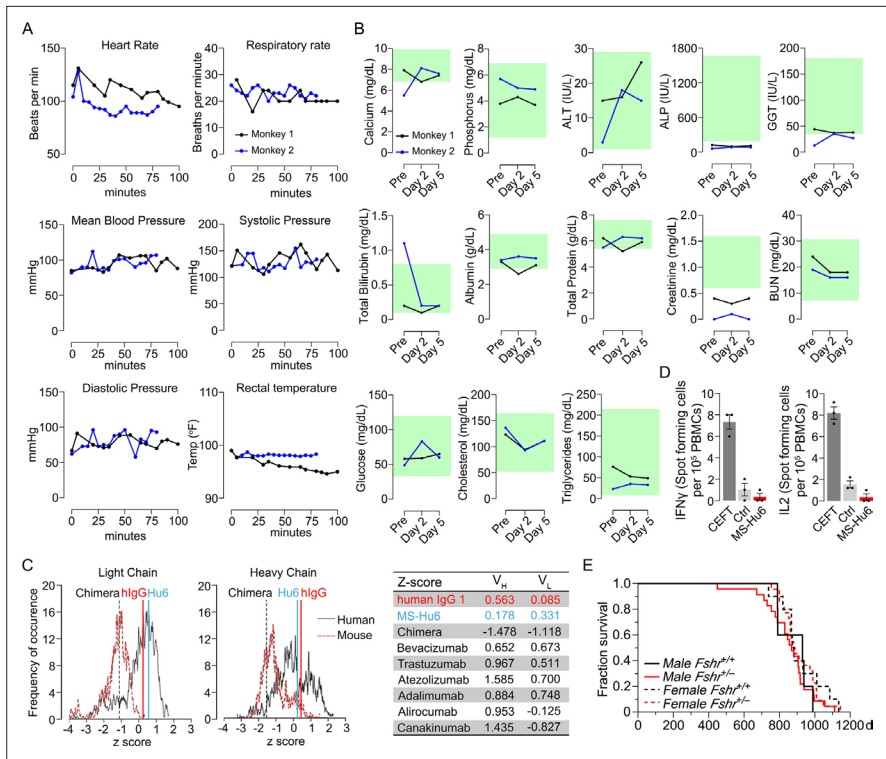

**Figure 4.** Acute and chronic tolerability of MS-Hu6. Effects on physiological parameters monitored up to 100 min (**A**) and serum biochemistry (at days 0, 2, and 5) (**B**) after injecting [89]Zr-MS-Hu6 as a single i.v. bolus dose (1.3 mg, ~1.3 mCi) into the tail veins of *Cynomolgus* monkeys ("Andy", 14 years/9.8 kg, and "Scott", 15 years/6.15 kg). Normative serum biochemistry data (green) from *Koo et al., 2019*. In silico assessment of "humanness" using abYsis, with Z-scores comparing humanized MS-Hu6, mouse-human chimeric antibody (*Gera et al., 2020*), and human IgG1 (**C**). Table shows narrowly distributed Z-scores after inputting primary sequences of FDA-approved, clinically utilized humanized ("zumab"), and fully human ("mab") antibodies: bevacizumab (anti-VEGF, Avastin); trastuzumab (anti-HER2, Herceptin); atezolizumab (anti-PD-L1, Tecentriq); adalimumab (anti-TNFα, Humira); alirocumab (anti-PCSK9, Praluent); and canakinumab (anti-IL-β; Ilaris) (**C**). Assessment of immunogenicity using ELISpot assays for the inflammatory cytokines IFNγ and IL-2 in human peripheral blood mononuclear cell cultures in response to added MS-Hu6, Dulbecco's Modified Eagle's Medium (DMEM) (Ctrl), or peptide pool from cytomegalovirus, Epstein-Barr virus, influenza and tetanus toxoid (CEFT) (positive control, Immunospot) (**D**). Kaplan–Meier survival curves showing that haploinsufficiency of *Fshr*, which otherwise mimics the effect of MS-Hu6 on bone mass (*Sun et al., 2006*; *Liu et al., 2017*), does not reduce lifespan compared with wild-type littermates over 1100 days (3 years; 5, 10 mice for wild-type male and female, and 23, 22 mice for male and female *Fshr+/-* mice, respectively) (**E**).

The online version of this article includes the following source data for figure 4:

**Source data 1.** Source data for *Figure 4*.

---

evidence, using our *Fshr+/-* mouse, that haploinsufficiency of the FSHR—which mimics the effect of FSH blockade on osteoporosis (*Sun et al., 2006*; *Ji et al., 2018*; *Liu et al., 2017*)—does not affect lifespan negatively in male or female mice (*Figure 4E*, *Figure 4—source data 1*).

## Developability, formulation, and physicochemistry of MS-Hu6

Therapeutic antibodies selected on the basis of affinity, potency, specificity, functionality, and pharmacokinetics should not have unsuitable physicochemical attributes making it difficult to streamline optimal manufacturing. It is therefore imperative that, at an early stage, we determine physicochemical properties, after a rough in silico check for 'red flags' (*Hebditch and Warwicker, 2017*; *Hebditch and Warwicker, 2019*; *Jain et al., 2017*). We thus used a computational tool, Protein–Sol Abpred, based on machine learning of amino acid sequences and physicochemical variables from 48 FDA-approved antibodies and 89 antibodies in late-stage clinical development (https://protein-sol.manchester.ac.uk/abpred) (*Hebditch and Warwicker, 2017*; *Hebditch and Warwicker, 2019*; *Jain et al., 2017*).

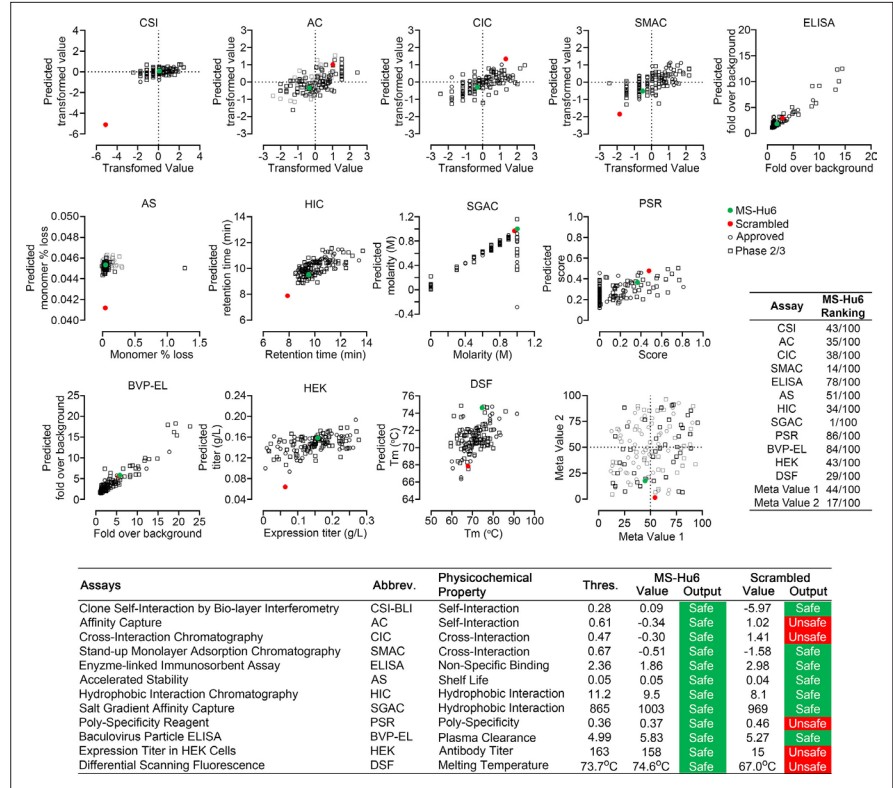

**Figure 5.** Manufacturability of MS-Hu6. Protein–Sol Abpred was used to compare twelve physicochemical parameters that were computationally derived for MS-Hu6 against experimental derivations of 48 FDA-approved antibodies and 89 antibodies in late-stage (phase 2/3) clinical development. MS-Hu6 fell within the "safe" range and was therefore considered manufacturable. The relative ranking of specific parameters is shown (1 = best, 100 = worst; Table, insert). A variation of MS-Hu6, in which the CDR regions were scrambled, failed 5 of 12 outputs, indicating non-manufacturability. While 65% FDA-approved monoclonal antibodies show no 'red flags,' those with up to four red flags have been approved by the FDA (*Jain et al., 2017*). Also shown are meta values for both MS-Hu6 and its scrambled sequence (1 = best; 100 = worst) derived by averaging ranks for eight experimental parameters. MS-Hu6 fell within the lower left quadrant, confirming that physicochemical properties were acceptable for manufacturing.

The online version of this article includes the following source data for figure 5:

**Source data 1.** Source data for *Figure 5*.

In the initial iteration, Protein–Sol Abpred provided predicted values for 12 separate physicochemical parameters that determine manufacturability. For all outputs, after inputting $V_H$ and $V_L$ regions, MS-Hu6 fell within acceptable thresholds and was therefore deemed to be "safe" (*Figure 5*, *Figure 5—source data 1*). In essence, the physicochemical properties of MS-Hu6 were likely to be broadly similar to FDA-approved antibodies.

For validation, we inputted a version of MS-Hu6, wherein the CDR region was scrambled—5 of 12 outputs, namely affinity capture (AC), cross-interaction chromatography (CIC), polyspecificity reagent (PSR), expression titers in HEK cells (HEK), and differential scanning fluorescence (DSF), fell outside the respective thresholds—an early indication that the scrambled version was not manufacturable (*Figure 5*). In fact, while 65% FDA-approved monoclonal antibodies show no 'red flags,' those with up to four red flags have been FDA-approved (*Jain et al., 2017*). To complement data from individual outputs, we derived a meta value for both MS-Hu6 and its scrambled sequence (1 = best; 100 = worst) by averaging ranks for eight experimental parameters (also shown in Table insert in *Figure 5*). We found that meta value pairs fell within the lower left quadrant, suggesting overall acceptable physicochemical properties, even in comparison with certain FDA-approved antibodies in the upper right quadrant (*Figure 5*, *Figure 5—source data 1*).

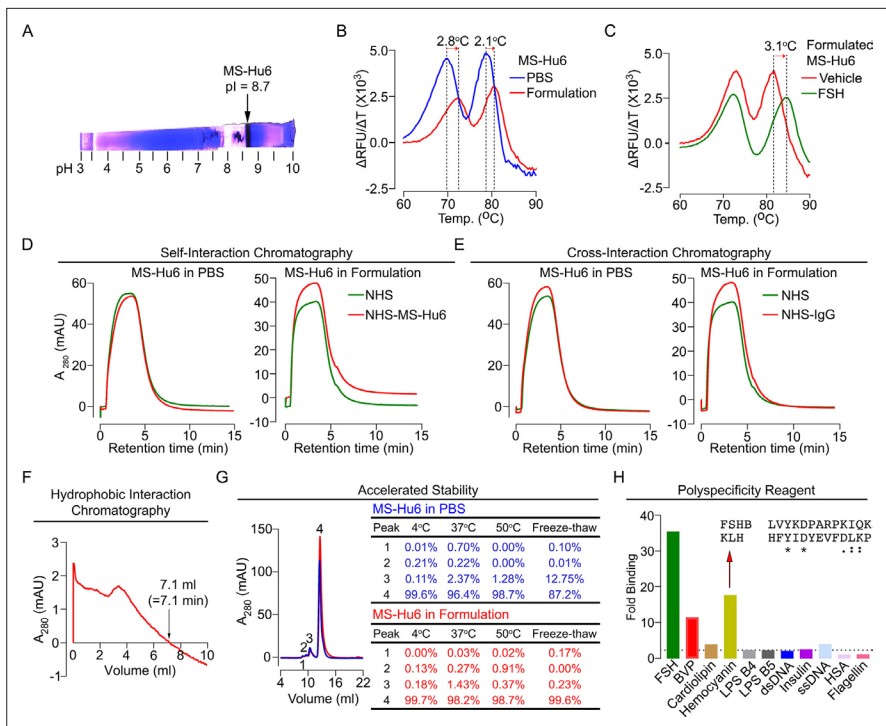

**Figure 6.** Physicochemical characteristics of MS-Hu6. Isoelectric focusing confirmed an isoelectric pH (pI) pf 8.7 for MS-Hu6, consistent with its in silico prediction (Expasy) of 8.58 (**A**). Thermal shift assays were used to examine the stability of both Fc and Fab regions of formulated MS-Hu6 versus MS-Hu6 in PBS (**B**), as well as the stability of follicle-stimulating hormone (FSH) binding to the Fab region of formulated MS-Hu6 (**C**). UV absorbance (280 nm) readout of self-interaction chromatography (SIC) (**D**) or cross-interaction chromatography (CIC) (**E**) to assess binding of formulated MS-Hu6 versus MS-Hu6 in PBS with self or human IgG, respectively. Hydrophobic chromatography showing UV absorbance (280 nm) of the eluate from a butyl sepharose column upon passing MS-Hu6 at pH 6.5 over 20 min at a flow rate of 1 mL/min (retention time shown) (**F**). Representative size exclusion chromatograms and area under the peak for MS-Hu6 in PBS or formulated MS-Hu6 following stress testing by three cycles of freeze-thaw or incubation at 4°C, 37°C, and 50°C for 1 week (**G**). Reactivity of MS-Hu6 to a standard panel of antigens, including cardiolipin, hemocyanin (KLH), lipopolysaccharides B4 and B5 (LPS B4 and LPS B5), double- and single- stranded DNA (dsDNA and ssDNA), insulin, human albumin, flagellin, and baculovirus particles (BVP) (ELISAs) (**H**). Homology of KLH with the epitope against which MS-Hu6 was raised, showing identical (*) and conserved (:) amino acid residues (**H**).

Before testing the physicochemical characteristics of MS-Hu6 experimentally, we created an optimal formulation. To prevent deamidation and isomerization at neutral and basic pHs, therapeutic antibodies are generally formulated at pHs away from their isoelectric pH (pI) (*Goswami et al., 2013*; *Wang et al., 2007*). Using Expasy, we predicted the pI for MS-Hu6 as 8.58. Isoelectric focusing confirmed a pI of 8.7 (*Figure 6A*). We tested 217 combinations of salt, detergent, and sugars for their thermal stability (not shown). This yielded in a near-final, research-grade formulation for MS-Hu6—stock solution of 2 mg/mL in 20 mM phosphate, 0.001% (v/v) Tween-20, 1 mM NaCl, and 260 mM sucrose (pH = 6.58). We found that both Fc and Fab regions of the formulated MS-Hu6 showed a thermal shift ($\Delta T_m$) compared with MS-Hu6 in PBS—and, hence, was confirmed as being more stable (*Figure 6B*). We also examined the binding of formulated MS-Hu6 to purified human FSH. A $\Delta T_m$ of 3.1°C of the Fab, and not the Fc region, established greater thermal stability due to FSH binding (*Figure 6C*).

For a full physicochemical characterization of the formulated MS-Hu6, we used a battery of biochemical tests, namely CIC, self–interaction chromatography (SIC), size exclusion chromatography (SEC), hydrophobic chromatography, and polyspecificity assays. For CIC and SIC, we created an in-house NHS ester column conjugated either with human IgG (for CIC) or MS-Hu6 (for SIC). MS-Hu6 was passed through an unconjugated column both in PBS and as a formulation, as well as through the two conjugated columns. *Figure 6D and E* show that retention times of MS-Hu6 in the respected

conjugated columns were not different from those of the unconjugated column—this confirms that MS-Hu6 does not appreciably interact with itself or with human IgGs, also indicative of little or no aggregation.

Highly soluble and hydrophilic monoclonal antibodies are expected to behave robustly during the manufacturing process. On the other hand, hydrophobic antibodies with high sensitivity to salt may display problems, such as poor expression, aggregation, or precipitation during purification. Delayed retention times in a chromatographic assay with a hydrophobic matrix are indicative of a tendency for precipitation. We used a butyl sepharose column and passed MS-Hu6 in 1.8 M $(NH_4)_2SO_4$ and 0.1 M $Na_3PO_4$ at pH 6.5 over 20 min at a flow rate of 1 mL/min. The ultraviolet absorbance was monitored at 280 nm to yield a retention time of 7.1 min (*Figure 6F*), which is below the theoretical threshold of 11.2 min (*Figure 5*).

We stressed formulated or PBS-containing MS-Hu6 through three freeze-thaw cycles (from –80°C to room temperature), and by incubation for 1 week at 4°C, 37°C, or 50°C, followed, in all cases, by SEC (*Figure 6G*). We noted a major peak (#4) and three minor high molecular weight peaks (#1–3). Areas under peaks 1 and 2 were between 0 and 0.91% of the total eluate under all conditions for both PBS-containing and formulated MS-Hu6. Peak 3 remained generally low (<1.5%) with formulated MS-Hu6 but was considerably higher (up to 12.7%) during freeze-thaw in PBS. Furthermore, the major peak 4 was consistently >99% with formulated MS-Hu6, particularly when compared to PBS-containing MS-Hu6 under freeze-thaw (87.2%). In all, the formulation protected against aggregation even under extreme stress conditions. Extending the protocol to 25 mL elution yielded no fragment peaks under any condition (*Figure 6G*).

Therapeutic antibodies must undergo a test for polyreactivity to a panel of relevant antigens, including cardiolipin, hemocyanin (KLH), lipopolysaccharide (LPS), double- and single-stranded DNA, insulin, human albumin, flagellin, and baculovirus particles (BVP). The fold change for MS-Hu6 binding to these antigens in ELISAs, except hemocyanin and BVP, was below a threshold of 3, and was therefore acceptable (*Figure 6H*). The BVP scores greater than 5 have been shown to enhance antibody clearance, with potential effects on $t_{1/2}$. In contrast, binding to hemocyanin, an arthropod protein, often used as a carrier for synthetic peptides during immunization (*Lateef et al., 2007*), was likely due to its homology with the 13-amino-acid sequence of FSHβ that binds MS-Hu6 (*Figure 6H*). However, as hemocyanin was not used during the production of the parent monoclonal antibody, Hf2, this is an irrelevant finding. This KLH-binding property is reflected in the low ranking of PSR (86/100) in the Protein–Sol, as well as BVP-ELISA (BVP-EL, 84/100; *Figure 5*).

## Discussion

Not many biologics, or indeed new therapeutics, undergo robust preclinical evaluations in academic medical centers outside of the pharmaceutical or biotechnology space. Our comprehensive analysis of the biology and physicochemistry of a first-in-class, humanized FSH-blocking antibody, MS-Hu6, establishing it as efficacious, safe, and manufacturable may arguably be amongst the first attempts at an extensive in-house effort. In 2003, we conceived the idea that pituitary hormones, including FSH, have direct actions on the bone (*Sun et al., 2006*; *Abe et al., 2003*; *Sun et al., 2019*; *Tamma et al., 2009*); shifted this new paradigm to establish novel actions of FSH on fat, and more recently, on the brain (*Liu et al., 2017*; *Xiong et al., 2022*); created the first polyclonal FSH-blocking antibody targeted to a 13-amino-acid-long receptor-binding epitope of FSHβ (*Zhu et al., 2012a*; *Zhu et al., 2012b*); and finally developed a humanized multipurpose therapeutic (*Gera et al., 2020*) for future use in osteoporosis, obesity, and perhaps even in hyperlipidemia and Alzheimer's disease (AD). Notably, the in-house studies described here have been carried out using our own platform that complies with Good Laboratory Practice, as mandated by the Food and Drug Administration's Code of Federal Regulations [Title 21, Chapter 1 A, Part 58]. This means that this Investigational New Drug-enabling dataset forms the basis of late-stage development and future first-in-human studies.

It is now well recognized that targeting the receptor-binding epitope of FSHβ to block its interaction with the FSHR increases bone mass, reduces body fat, enhances thermogenesis, and prevents neurodegeneration in mice (*Ji et al., 2018*; *Liu et al., 2017*; *Zhu et al., 2012a*; *Zhu et al., 2012b*; *Geng et al., 2013*; *Han et al., 2020*). Notably, certain of these effects are triggered not only by our polyclonal and monoclonal FSH-blocking antibodies (*Ji et al., 2018*; *Liu et al., 2017*; *Zhu et al., 2012a*; *Zhu et al., 2012b*) but also through the use of vaccines, such as a GST-FSHβ fusion protein

(*Geng et al., 2013*) or tandem repeats of the epitope (*Han et al., 2020*). It is important to note that even with high antibody doses, such as 200 μg/mouse/day, serum estrogen levels are unchanged (*Liu et al., 2017*; *Zhu et al., 2012a*), likely because of the abundance of FSHRs in the ovary that remains responsive to lowered levels of bioactive (unblocked) FSH. These preclinical data in mice establishing FSH as an actionable target are reinforced by striking estrogen-independent correlations in women between serum FSH, rapid bone loss, visceral obesity, and spikes of cognitive decline during the menopausal transition, at which time serum estrogen is relatively normal and FSH levels are rising (*Sowers et al., 2001*; *Sowers et al., 2007*; *Sowers et al., 2006b*; *Randolph et al., 2004*; *Randolph et al., 2011*; *Randolph et al., 2003*). This window is likely the most opportune to prevent bone loss, obesity, and early cognitive decline through selective FSH blockade. In this regard, while negative data with GnRH modulators are mostly confounded by concomitant changes in LH and GnRH (*Drake et al., 2010*), it is clear that low gonadotropin levels in triptorelin-treated men with prostate cancer are associated with lower fat mass and body weight than men undergoing orchiectomy, where gonadotropins are high (*Østergren et al., 2019*). Overall, therefore, the data together lend credibility to the idea that FSH-driven changes, at the very least in body composition, in both sexes can be rescued by blocking FSH. The selective inhibition of FSH action, therefore, becomes a worthy imperative.

Menopause is also associated with dyslipidemia, which has long been thought to result from estrogen deficiency. However, there is compelling epidemiologic evidence that high serum FSH levels correlate with serum total and LDL cholesterol in post-menopausal women, and importantly, that total cholesterol rises across the perimenopausal transition, essentially tracking closely with bone loss and obesity (*Sowers et al., 2007*; *Sowers et al., 2003a*; *Song et al., 2016*; *van Beresteijn et al., 1993*). Impressively, exogenous FSH, in the presence of estrogen clamped at normal levels, increases serum total cholesterol in mice fed on a high cholesterol diet (*Guo et al., 2019*). And, consistent with the idea that FSH is an estrogen-independent driver of menopausal hypercholesteremia, an FSH-blocking antibody lowers serum cholesterol (*Zhu et al., 2012b*; *Guo et al., 2019*). There is also human evidence that reducing serum FSH by >30% from its zenith in post-menopausal women through estrogen replacement therapy lowers serum cholesterol (*Song et al., 2016*). As in the case of bone and fat cells (*Sun et al., 2006*; *Liu et al., 2017*), hepatocyte FSHRs couple with $Gi_{2\alpha}$, which signals through Akt to inhibit FoxO1 binding with the *Srebf2* promoter and prevent its repression. Upregulated *Srebf2*, which drives de novo cholesterol biosynthesis, results in increased cholesterol accumulation and release (*Guo et al., 2019*). This action is in addition to the lowering of LDLR expression by FSH (*Song et al., 2016*). Notwithstanding cholesterol-lowering mechanism(s), which are likely to be explored even further, it is possible that an FSH-blocking therapeutic, such as MS-Hu6, could have additional actions on lipid metabolism in people.

Furthermore, what underpins the preponderance of AD in post-menopausal women, particularly in relation to disease risk, progression, and severity, has remained unclear, now for decades. A role for post-menopausal hypoestrogenemia remains controversial, with improvement (*Matyi et al., 2019*), no change (*Zandi et al., 2002*; *O'Brien et al., 2014*), or worsening (*Shumaker et al., 2004*) of cognition with estrogen replacement therapy. In contrast, high serum FSH is strongly associated with the onset of AD and has thus been suggested as a possible mediator (*Short et al., 2001*; *Bowen et al., 2000*). More importantly, certain neuropathologic features, including neuritic plaques, neurofibrillary tangles, and gliosis often begin during the perimenopausal transition (*Hampel et al., 2018*; *Dubois et al., 2016*; *Jack et al., 2013*; *Epperson et al., 2013*; *Greendale et al., 2009*). During this period, women also show a sharp decline in memory function and an increased risk of mild cognitive impairment and dementia (*Randolph et al., 2011*; *Epperson et al., 2013*; *Greendale et al., 2009*). We have recently documented exaggerated AD pathology and cognitive decline upon ovariectomy or exogenous FSH injection in three murine models of AD, even in the face of estrogen levels clamped in the normal range (*Xiong et al., 2022*). This phenotype arises from the action of FSH on hippocampal and cortical neuronal receptors using a pathway involving the transcription factor CCAAT enhancer-binding protein beta and the δ-secretase asparagine endopeptidase (*Xiong et al., 2022*). Most notably, however, we found that our polyclonal FSH-blocking antibody, which shares the target epitope with MS-Hu6 (*Zhu et al., 2012a*; *Gera et al., 2020*), prevented the AD-like phenotype induced upon ovariectomy (*Xiong et al., 2022*)—providing a clear avenue for further exploration of the effects of MS-Hu6 in models of AD.

In all, therefore, we and others have unraveled new actions of FSH that assign it as an actionable target requiring a highly specific approach to block its action in people. We believe that MS-Hu6 with an affinity approaching that of trastuzumab (*Genentech, 1998*) is poised for future testing in human trials. Admittedly ambitious, we envisage, in a best-case scenario, and if mouse data translate into people, of treating four diseases that affect millions of women and men worldwide—namely obesity, osteoporosis, dyslipidemia, and neurodegeneration—with a single multipurpose FSH-blocking agent.

## Methods

### Animals

Colonies of male and female C57BL/6 mice and male Tg32, *Fshr*[+/-], and ThermoMice were obtained from Jackson Labs. Male CD1 mice were from Charles River Laboratories. Mice were maintained in-house at Icahn School of Medicine at Mount Sinai and/or Maine Medical Center Research Institute. They were either fed on normal chow or on a high-fat diet (DIO Formula D12492, 60% fat; Research Diets, Inc, New Brunswick, NJ.), with access to water ad libitum. The mice were housed in climate-controlled conditions with standard 12-hour light/dark cycles (6 AM to 6 PM). Nonhuman primates, namely *Cynomolgus* monkeys (*Macaca fascicularis*), were fed Teklad Global 20% Protein Primate Diet. All protocols were approved by the Institutional Animal Care and Use Committees of Icahn School of Medicine at Mount Sinai and Maine Medical Center Research Institute.

### Bone phenotyping

At Maine Medical Center Research Institute, BMD was measured in C57BL/6 and C3H/HeJ mice post-ovariectomy and MS-Hu6 treatment by dual-energy X-ray absorptiometry (*PIXImus*, Lunar) with a precision of <1.5% (*Sun et al., 2003*). Anesthetized mice were subject to measurements, with the cranium excluded. The instruments were calibrated each time before use per the manufacturer's recommendation. For μCT measurements at USDA (North Dakota), femoral epiphyses were scanned non-destructively by using a Scanco μCT scanner (μCT-40; Scanco Medical AG, Bassersdorf, Switzerland) at 12 μm isotropic voxel size, with X-ray source power of 55 kV and 145 μA, and integration time of 300 ms. The trabecular microstructure was evaluated after removing the noise from the scanned grayscale images using a low-pass Gaussian filter. A fixed BMD threshold of 220 mg/cm$^3$ was used to extract the mineralized bone from soft tissue and the marrow phase. Reconstruction and 3D quantitative analyses were performed using software provided by Scanco. The same settings for scan and analysis were used for all samples. Trabecular bone parameters included fractional bone volume (BV/TV), Tb.Th, Tb.N, Tb.Sp, and Conn.D. The cortical bone parameter included Ct.Th.

The 2D histomorphometry was performed in femoral epiphysis and spine (L1-L3). Frozen non-decalcified sections (6–8 μm) were stained with a von Kossa staining kit (American MasterTech, catalog # KTVKO), per the manufacturer's procedure. This provided measures of fractional bone volume (B.Ar/T.Ar), Tb.N, Tb.Sp, and Tb.Th. Bone formation was quantified by dynamic histomorphometry following sequential injections of calcein (15 mg/kg) followed by xylelol orange (90 mg/kg) 8 days apart, with the last injection 4 days prior to sacrifice. Parameters included mineralizing surface (MS), MAR, and BFR. Osteoclast resorbed surface (Oc.S/BS) was measured following TRAP staining with Leukocyte Acid Phosphatase TRAP kit (Sigma Aldrich, catalog # 387A-1KT).

For hormone measurements, we used the following ELISA kits: FSH and LH (Milliplex MAP Mouse Pituitary Magnetic Beads, catalog #: MPTMAG-49K, CLHMAG, and RFSHMAG, respectively), total inhibin (R&D Systems, catalog #: LXSAHM), and activin A (ThermoFisher, catalog #: EM3RB).

### Pharmacokinetics and biodistribution in mice

We prepared [89]Zr-MS-Hu6 by incubating MS-Hu6 first with the chelator DFO-p-NCS for 3 hr at 37°C (in steps of 5 μL until a tenfold molar excess of chelator was achieved). The DFO-functionalized MS-Hu6 was washed three times with PBS in a 10 kDa-cutoff ultrafiltration tube before radiolabeling. [89]Zr-oxalate was diluted with PBS and neutralized with 1 M $Na_2SO_4$ before adding to functionalized MS-Hu6. This was followed by incubation with [89]Zr-oxalate for 1 hr at 37°C, ultrafiltration (10 kDa cut-off), and thin layer chromatography (with 50 mM EDTA) for quality check (*van de Watering et al., 2014*). C57BL/6 or Tg32 mice were injected in separate experiments with [89]Zr-MS-Hu6 as a single dose of ~250 μCi (~250 μg, 250±40 μCi) into the retro-orbital sinus. Timed blood (few drops drawn from

the tail vein) and excreta collection was followed by weighing and γ-counting (Wizard 2480 Automatic Gamma Counter, Perkin Elmer, Waltham, MA). Values were corrected for decay and expressed as a percentage of injected dose per gram of tissue. Radiolabeling (using 1.0 μCi per μg of MS-Hu6) was performed the afternoon before the day of injections. At the time of injection, the activity:antibody ratio was likely closer to 0.85 μCi/μg. The $C_{max}$ values using $^{89}$Zr-MS-Hu6 are therefore rough estimates.

In complementary experiments, MS-Hu6 was biotinylated by incubating with NHS ester-biotin (100 μg per mg MS-Hu6, dissolved in DMSO, Sigma, catalog # H1759) for 4 hr at room temperature. NaHCO$_3$ (1 M) was then added for 10 min at room temperature (pH 8) and the product was purified through ultrafiltration (10 kDa cut-off). Biotinylated MS-Hu6 (200 μg) was injected i.p. into CD1 mice, which at timed intervals underwent cardiac puncture for blood sampling and sacrifice. We used an in-house ELISA in which the plate was coated with individual sera and biotinylated MS-Hu6 was captured by streptavidin-HRP (Millipore, catalog # 18–152). In a final experiment, unconjugated MS-Hu6 was injected i.p. into Tg32 mice (200 μg). Serum was collected from groups of mice and an in-house assay was used in which anti-human IgG Fc (Sigma, catalog # FAB3700259) was used to capture human IgG (MS-Hu6), with detection of the complex with goat anti-human IgG Fab (200 μg/well; Invitrogen, #31122). For all pharmacokinetics studies, $C_{max}$ and β phase $t_{1/2}$ were estimated using a two-phase decay curve fitting in Prism v.9.4.1 (Graphpad). For *Figures 2D and 3* aberrantly low data points, which were 2.5 standard deviations below the group mean, were excluded as they falsely elevated β phase $t_{1/2}$ to 316 hr. The data was refitted to yield a β phase $t_{1/2}$ of 180 hr.

For imaging, $^{89}$Zr-MS-Hu6-treated mice (above) were anesthetized using 1% isoflurane in O$_2$ at a flow rate of ~1.0 L/min. The PET/CT scans were performed using a Mediso nanoScan PET/CT (Mediso, Budapest, Hungary). For whole body CT scans, we used the following parameters: energy, 50 kVp; current, 180 μAs; and isotropic voxel size, 0.25 mm—this was followed by a 30-min PET scan. Image reconstruction was performed with attenuation correction using the TeraTomo 3D reconstruction algorithm from the Mediso Nucline software (version 3.04). The coincidences were filtered with an energy window between 400 and 600 keV. Voxel size was isotropic with 0.4 mm width, and the reconstruction was applied for four full iterations, six subsets per iteration. Image analysis was performed using Osirix MD, version 11.0. Namely, whole body CT images were fused with PET images and analyzed in an axial plane. The ROIs were drawn on various tissues. Testis, visceral WAT, subcutaneous WAT, kidneys, liver, and brain were traced in their entirety, and bone marrow uptake was assessed using three vertebrae in the lumbar spine. Mean SUVs (normalized to muscle) were calculated for each ROI. Subsequently, $^{89}$Zr-MS-Hu6 uptake of each tissue was expressed as the average of all mean SUV values per organ. After imaging, the mice were sacrificed and perfused with 20 mL of PBS and tissues of interest, namely brain, heart, kidney, pancreas, liver, lung, bone, bone marrow, BAT, subcutaneous WAT, visceral WAT, adrenal, blood, testis, spleen, and muscle, were isolated for γ-counting.

For biodistribution studies using AF750-labeled MS-Hu6 (Alexa Fluor 750, Invitrogen, catalog # S30046; SAIVI Rapid Antibody Labeling kit; human IgG1 as negative control [Sigma catalog # I12511]), we first imaged the whole body 16 hr after injection using IVIS Spectrum in vivo imaging system (Perkin Elmer). Mice were then perfused with PBS, sacrificed and organs, namely heart, thymus, brain, gastrocnemius and soleus muscle, BAT, adrenals, liver, gall bladder, spleen, kidney, subcutaneous WAT, visceral WAT gonadal WAT, testes, and bone were removed and imaged using the same IVIS platform to calculate average radiance efficiency per square area. Fluorescence in sections was examined using AF488 conjugated with Alexa Fluor 488 Antibody Labelling kit [Invitrogen, catalog # A20181] to MS-Hu6.

## Biodistribution and safety studies in monkeys

After an overnight fast, two male *Cynomolgus* monkeys ("Scott", aged 14 years, weight 9.8 kg; and "Andy", aged 15 years, weight 6.15 kg) were anesthetized with ketamine (5.0 mg/kg) and dexmedetomidine (0.75–1.5 μg/kg). The monkeys were injected with $^{89}$Zr-MS-Hu6, and blood was drawn at 30 min, and at 48 and 120 hr from the tail vein. Vitals, including mean arterial, systolic and diastolic blood pressure, respiratory rate, heart rate, and rectal temperature, were recorded using the Waveline Touch system (DRE) and Welch Allyn rectal thermometer. The PET and MR images were acquired on a combined 3T PET/MRI system (Biograph mMR, Siemens Healthineers). Whole body MR images from each PET bed (head, thorax, and pelvis) were automatically collated together with a scanner. The MR parameters were as follows: acquisition plane, coronal; repetition time, 1000 ms; echo time, 79

ms; number of slices, 224; number of average, 2; spatial resolution of 0.6 mm × 0.6 mm × 1.0 mm; acquisition duration, 29 min; and 56 s per bed. After the acquisition, PET raw data from each bed were reconstructed and collated offline using the Siemens proprietary e7tools with an ordered subset expectation maximization algorithm with point spread function correction. A dual-compartment (soft tissue and air) attenuation map was used for attenuation. Image analysis was performed using Osirix MD, version 11.0. Whole-body MR images were fused with PET images and analyzed in an axial plane. The ROIs were drawn on various tissues. The liver, kidney, BAT (interscapular region), subcutaneous WAT, visceral WAT, gonadal WAT, gallbladder, spleen, brain, and testes were traced in their entirety; bone marrow was imaged from the shoulder; and three lumbar vertebrae; and muscle was imaged from the quadriceps. Mean SUVs were calculated for each ROI. $^{89}$Zr-MS-Hu6 uptake of each tissue was expressed as the average of all mean SUV values per organ. Serum was collected for blood chemistry analysis by IDEXX BioAnalytics.

## ELISpot assay

Human peripheral blood mononuclear cells (obtained from Immunospot, Cellular Technology Ltd.) were cultured for 12 days in FBS-free DMEM with regular medium change, and plated at a density of $10^5$ cells/well in ImmunoSpot ELISpot plates. Cells were then exposed to MS-Hu6, CEFT, or DMEM for 48 hr, following which IL-2 and IFN-γ expressing cells were quantitated per the manufacturer's instructions.

## In silico analyses

We used Protein–Sol Abpred, a computational algorithm based on machine learning of amino acid sequences and physicochemical variables from 136 antibodies that are FDA-approved or in late-stage clinical development (https://protein-sol.manchester.ac.uk/abpred; *Hebditch and Warwicker, 2017*; *Hebditch and Warwicker, 2019*; *Jain et al., 2017*). Protein–Sol Abpred uses antibody sequences ($V_H$ and $V_L$) as inputs to provide predicted outputs for clone self-interaction by bio-layer interferometry (CSI-BLI), PSR, BVP-EL, CIC, ELISA, accelerated stability, hydrophobic interaction chromatography (HIC), standup monolayer adsorption chromatography, salt gradient affinity capture, expression titer in HEK cells (HEK), AC, and DSF. It also provides a meta value for each sequence (1 = best; 100 = worst) by averaging ranks for eight experimental parameters. For predicting the pI value of MS-Hu6, we inputted its sequence into Expasy (Swiss Bioinformatics Research Portal, https://web.expasy.org/compute_pi/). For predicting the "humanness" of humanized MS-Hu6 versus parent chimera and other humanized or fully human molecules (*Gera et al., 2020*), we inputted sequences into abYsis (version 3.4.1; http://www.abysis.org/abysis/sequence_input/key_annotation/key_annotation.cgi) to obtain Z-scores, and compared the scores with that of $V_H$ and $V_{LK}$ chain of IgG1.

## Protein thermal shift assay

The protein thermal shift assay used a fluorescent reporter, Sypro-Orange (Protein Thermal Shift Dye kit, ThermoFisher, catalog # 4461146), to detect hydrophobic domains that are exposed following the heat-induced unfolding of globular proteins. MS-Hu6 (1.5 µg/µL), formulated or in PBS, was incubated with or without human FSH (0.5 µg/µL) at room temperature for 30 min, with fluorescence captured sequentially at 0.3°C increments using a StepOne Plus Thermocycler (Applied Biosystems). The $T_m$ was calculated based on the inflection point of the melt curve, and the thermal shift was derived from $\Delta T_m = T_mA - T_mB$.

## Isoelectric focusing

For determining the pI, 2D electrophoresis was performed by first rehydrating MS-Hu6 (500 µg) for 2 hr at room temperature in rehydration buffer (8 M urea, 2% CHAPS, 0.5% IPG buffer, and trace of bromophenol blue) without DTT. The sample was then run on an 18 cm 3–10 strip using the Ettan IPGphor 3 Isoelectric Focusing system (GE Healthcare). Four voltage steps (50 V for 10 hr; 500 V for 1 hr; 1000 V for 1 hr; 8000 V for 4 hr) were followed by Coomassie blue staining.

## Chromatography

An NHS ester column (HiTrap NHS Activated HP, Cytiva, catalog # 17071601) was conjugated with either human IgG (for CIC) or MS-Hu6 (for SIC) in 0.2 M NaHCO₃, 0.5 M NaCl (pH 8), per manufacturer.

Unbound IgG was removed by three successive washings at a rate of 0.4 mL/min using Buffer A (0.5 M ethanolamine, 0.5 M NaCl, pH 8.3) and Buffer B (0.1 M sodium acetate, 0.5 M NaCl, pH 4). An unconjugated column was prepared without including any IgG in the coupling step. MS-Hu6, either as a formulation or in PBS, was run through the unconjugated column, followed by either conjugated columns at 0.2 mL/min. For SEC, we used a Superdex 200 Increase 10/300 GL (GE Lifesciences, catalog # 28990944) and passed MS-Hu6 (2 mg) in either PBS or formulation buffer after three cycles of freeze-thaw or incubation for 1 week at 4°C, 37°C, or 50°C. For HIC, we used HiTrap Butyl FF Sepharose column (GE Lifesciences, catalog # GE17-1357-01) to run MS-Hu6 at 1 mL/min in a linear gradient from 1 M to 0 M $(NH_4)_2SO_4$ (generated using 1.8 M $(NH_4)_2SO_4$ and 0.1 M $Na_3PO_4$ [pH 6.5], and 0.1 M $Na_3PO_4$ [pH 6.5]). For CIC, SIC, SEC, and HIC, absorbance was monitored at 280 nm using the ÄKTA Pure FPLC (GE Lifesciences), and the data was analyzed by Unicorn version 6.4.

## Polyspecificity testing

We adapted a previous method to determine antibody polyspecificity (*Jain et al., 2017*) and developed an in-house ELISA by coating plates with multiple antigens (50 nM for each), namely double-stranded DNA (Shear Salmon Sperm DNA, 5'–3' Inc, catalog # 5302–754688), single-stranded DNA (Deoxyribonucleic acid, single stranded from Calf Thymus, Sigma, catalog # D8899), cardiolipin (Cardiolipin sodium salt from bovine heart, Sigma, catalog # C0563), LPS-B5 (LPS *Escherichia coli* 055:B5, Calbiochem, catalog # 437625), LPS-B4 (LPS-EB, InvivoGen, catalog # tlrl-eblps, 50 nM), KLH (keyhole limpet, Hemocyanin from *Megathura crenulata*, Sigma, catalog # H8283), insulin (Humulin R, Lilly, catalog # HI213), BVP (Medna, catalog #: E3001), flagellin (Adipogen, Catalog # AG-4013–0095 C100), human serum albumin (Sigma, catalog # A9511), and human pituitary FSH (National Hormone and Pituitary Program, UCLA). The antigens were exposed to MS-Hu6 (100 nM), overnight at 4°C, and any antigen-MS-Hu6 complex was captured by goat anti-human HRP-conjugated IgG (Invitrogen, catalog # A18805).

## Statistical methods

Statistically significant differences between any two groups were examined using a two-tailed Student's *t*-test, given equal variance. The p values were considered significant at or below 0.05.

### Ensuring rigor and reproducibility

There is a nascent movement to ensure that preclinical data is true and accurate (*Ioannidis, 2005*; *Collins and Tabak, 2014*; *McNutt, 2014*; *Mullard, 2017*; *Horrigan et al., 2017a*; *Horrigan et al., 2017b*). M.Z. and C.J.R. coined the phrase 'contemporaneous reproducibility,' which refers to the synchronous reproduction of data in more than one laboratory. As Zaidi's discovery of the effects of FSH on bone and body fat were novel, he reached out to C.J.R. for help in form of a reproducibility study. Key data sets were reproduced by C.J.R in a process that lasted over three years, as other validation studies were added by both laboratories. The term replicability refers to the ability of one or more independent groups to replicate a finding using a different technology or method—replicability is a measure of truth or significance of a given finding (*Rosen and Zaidi, 2017*). Here, we have replicated a key finding that Hu6 increased bone mass in the M.Z. and C.J.R. labs, with μCT data independently produced by J.J.C. To further enhance transparency, we have hosted detailed procedures and raw datasets on our GLP-compliant MediaLab Document Control System, which all investigators have access to. All data have undergone quality checks before the final product was signed off. Such practices requiring unfettered transparency remain fundamental to ensuring rigor.

## Acknowledgements

Work at Icahn School of Medicine at Mount Sinai performed at the Center for Translational Medicine and Pharmacology was supported by R01 AG071870 to MZ, TY and S-MK; R01 AG074092 and U01 AG073148 to TY and MZ; U19 AG060917 to MZ and CJR; and R01 DK113627 to MZ and JI. Work at U.S. Department of Agriculture, Agricultural Research Service, Grand Forks Human Nutrition Research Center (USDA ARS GFHNRC) was supported by the Project Plan #3062-51000-053-00D to JJC. Mention of trade names or commercial products in this publication is solely for the purpose of providing specific information and does not imply recommendation or endorsement by the U.S. Department of Agriculture. USDA is an equal opportunity provider and employer. The findings and

conclusions in this manuscript are those of the authors and should not be construed to represent any official USDA of U.S. Government determination or policy. MZ also thanks the Harrington Discovery Institute for the Innovator–Scholar Award towards development of the FSH antibody. CJR acknowledges support from the NIH (P20 GM121301).

## Additional information

### Competing interests

Jameel Iqbal, Tony Yuen: Reviewing editor, *eLife*. Mone Zaidi: is an inventor on issued patents on inhibiting FSH for the prevention and treatment of osteoporosis and obesity (U.S. Patent 8,435,948 and 11,034,761). M.Z. is also an inventor on pending patent application on composition and use of humanized monoclonal anti-FSH antibodies, and is co-inventor of a pending patent on the use of FSH as a target for preventing Alzheimer's disease. These patents are owned by Icahn School of Medicine at Mount Sinai (ISMMS), and M.Z. would be recipient of royalties, per institutional policy. M.Z. also consults for several financial platforms, including Gerson Lehman Group and Guidepoint, on drugs for osteoporosis and genetic bone diseases. Deputy editor, *eLife*. The other authors declare that no competing interests exist.

### Funding

| Funder | Grant reference number | Author |
| --- | --- | --- |
| National Institute on Aging | R01 AG071870 | Tony Yuen<br>Se-Min Kim<br>Mone Zaidi |
| National Institute on Aging | R01 AG074092 | Mone Zaidi<br>Tony Yuen |
| National Institute on Aging | U01 AG073148 | Mone Zaidi<br>Tony Yuen |
| National Institute on Aging | U19 AG060917 | Mone Zaidi<br>Clifford J Rosen |
| National Institute of Diabetes and Digestive and Kidney Diseases | R01 DK113627 | Mone Zaidi<br>Jameel Iqbal |
| National Institute of General Medical Sciences | P20 GM121301 | Clifford J Rosen |

The funders had no role in study design, data collection and interpretation, or the decision to submit the work for publication.

### Author contributions

Sakshi Gera, Conceptualization, Data curation, Formal analysis, Supervision, Validation, Investigation, Visualization, Methodology, Writing – original draft, Project administration, Writing – review and editing; Tan-Chun Kuo, Data curation, Formal analysis, Investigation, Visualization, Methodology, Writing – original draft, Writing – review and editing; Anisa Azatovna Gumerova, Vitaly Ryu, Data curation, Formal analysis, Investigation, Methodology; Funda Korkmaz, Data curation, Formal analysis, Validation, Investigation, Visualization, Methodology; Damini Sant, Formal analysis, Investigation, Visualization, Methodology; Victoria DeMambro, Data curation, Formal analysis, Supervision, Validation, Investigation, Visualization, Methodology; Karthyayani Sudha, Pushkar Kumar, Liam Cullen, Megha Bhongade, Data curation, Investigation; Ashley Padilla, Tomas GJM Post, Jessica C Fernandes, Jessica Netto, Farhath Sultana, Eleanor Shelly, Sari Miyashita, Hasni Kannangara, Kseniia Ievleva, Valeriia Muradova, Se-Min Kim, Data curation, Formal analysis, Investigation; Geoffrey Prevot, Data curation, Formal analysis, Investigation, Visualization, Methodology; Jazz Munitz, Data curation, Software, Formal analysis, Validation, Investigation, Visualization, Methodology; Abraham Teunissen, Data curation, Formal analysis, Supervision, Investigation, Visualization, Methodology; Mandy MT van Leent, Formal analysis, Supervision, Validation, Investigation, Methodology; Satish Rojekar, Anusha

Pallapati, Puja Sengupta, Investigation; Jiya Chatterjee, Data curation; Rogerio Batista, Data curation, Formal analysis; Cemre Robinson, Supervision, Investigation; Anne Macdonald, Maria I New, Zahi A Fayad, Daria Lizneva, Supervision, Project administration; Susan Hutchison, Data curation, Investigation, Visualization; Mansi Saxena, Marcia Meseck, Neeha Zaidi, Supervision, Methodology; John Caminis, Jameel Iqbal, Supervision, Methodology, Project administration; Jay J Cao, Data curation, Formal analysis, Methodology; Clifford J Rosen, Data curation, Formal analysis, Supervision, Funding acquisition, Validation, Investigation, Visualization, Methodology, Project administration; Tony Yuen, Data curation, Supervision, Funding acquisition, Validation, Writing – original draft, Project administration, Writing – review and editing; Mone Zaidi, Conceptualization, Supervision, Funding acquisition, Writing – original draft, Project administration, Writing – review and editing

## Author ORCIDs
Sakshi Gera ⓘ http://orcid.org/0000-0002-1615-6259
Tan-Chun Kuo ⓘ http://orcid.org/0000-0001-5301-755X
Abraham Teunissen ⓘ http://orcid.org/0000-0002-0401-8262
Vitaly Ryu ⓘ http://orcid.org/0000-0001-8068-4577
Clifford J Rosen ⓘ http://orcid.org/0000-0003-3436-8199
Mone Zaidi ⓘ http://orcid.org/0000-0001-5911-9522

## Ethics

This study was performed in strict accordance with the recommendations in the Guide for the Care and Use of Laboratory Animals of the National Institutes of Health. All of the animals were handled according to approved institutional animal care and use committee (IACUC) protocols (PROTO201900157) for non-human primate studies and (PROTO202100038) for mouse experiments at Icahn School of Medicine at Mount Sinai and at Maine Medical Center Research Institute.

## Decision letter and Author response

Decision letter https://doi.org/10.7554/eLife.78022.sa1
Author response https://doi.org/10.7554/eLife.78022.sa2

# Additional files

## Supplementary files
• Transparent reporting form

## Data availability
All data generated or analysed during this study are included in the manuscript and supporting file.

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
