## [Editor Report]

In this manuscript, the authors describe a comprehensive characterization of a new humanized FSH blocking antibody (MS-Hu6), which they have studied in depth in terms of its efficacy on bone. They provide compelling data on mouse and monkey species with a complete evaluation of their pharmacokinetics and biodistribution and characterize their effect on the treatment of obesity and bone loss.

---

## [Decision Letter]

**Decision letter after peer review:**

Thank you for submitting your article "A Single Multipurpose FSH-Blocking Therapeutic for Osteoporosis and Obesity" for consideration by *eLife*. Your article has been reviewed by 2 peer reviewers, one of whom is a member of our Board of Reviewing Editors, and the evaluation has been overseen by Carlos Isales as the Senior Editor. The reviewers have opted to remain anonymous.

Essential revisions:

1) While the authors document that endogenous FSH is bound to the antibody, they have not measured FSH levels after injecting antibody or IgG. In previous studies (Zhu et al., PNAS, and Ji et al., PNAS) they show no changes in detectable FSH after polyclonal and monoclonal antibodies, respectively. Is there a change with their humanized monoclonal? If so, an explanation is required.

2) The authors present that their FSH blocking antibody does not change in serum LH or testosterone in male mice, and show reduced activin. Have they measured inhibin levels? If possible, please provide.

3) The C3H bone data is interesting and serves as a good control for the responses in C57BL6 mice. Have the authors measured cortical bone at the femoral diaphysis? Please provide that data for both C57BL6 and C3H mice.

*Reviewer #1 (Recommendations for the authors):*

Overall, this is an elegant and complete analysis of a novel molecule with the potential of becoming a therapeutic. The level of comprehensiveness and rigor during preclinical development in this study is rare and is to be commended. The studies are well done with appropriate controls, and the authors have used multiple strategies to substantiate their findings. It is an important contribution and would be useful to a general readership in endocrinology, bone and fat metabolism.

*Reviewer #2 (Recommendations for the authors):*

This study is a well-designed set of experiments as a comprehensive characterization of MS-Hu6 for its efficacy in mouse models of obesity and osteoporosis, acute safety in monkeys, a full evaluation of its pharmacokinetic, pharmacodynamic and biodistribution, and a compendium of its physicochemical properties for treatment of obesity and bone loss in people. The manuscript is also well written. The manuscript is ready for publication.

---

## [Author Response]

Essential Revisions (for the authors):1) While the authors document that endogenous FSH is bound to the antibody, they have not measured FSH levels after injecting antibody or IgG. In previous studies (Zhu et al., PNAS, and Ji et al., PNAS) they show no changes in detectable FSH after polyclonal and monoclonal antibodies, respectively. Is there a change with their humanized monoclonal? If so, an explanation is required.2) The authors present that their FSH blocking antibody does not change in serum LH or testosterone in male mice, and show reduced activin. Have they measured inhibin levels? If possible, please provide.3) The C3H bone data is interesting and serves as a good control for the responses in C57BL6 mice. Have the authors measured cortical bone at the femoral diaphysis? Please provide that data for both C57BL6 and C3H mice.

The authors are very grateful to both reviewers for their critique and for the thorough and thoughtful review of our manuscript. We have made the requested changes, which includes new data. In addition, we provide new two–dimensional morphometry and dynamic histomorphometry data on the lumbar spine (L1-L3) in response to MS-Hu6 treatment. For the sake of focus, and in an effort to further validate and extend our body fat data in IND–enabling dose–finding studies, we have elected to remove this dataset and focus on the bone phenotype, in addition to the pharmacokinetics, biodistribution, tolerability, and manufacturability of MS-Hu6. This has reflected in a modification to the title.

Reviewer #1 (Recommendations for the authors):Overall, this is an elegant and complete analysis of a novel molecule with the potential of becoming a therapeutic. The level of comprehensiveness and rigor during preclinical development in this study is rare and is to be commended. The studies are well done with appropriate controls, and the authors have used multiple strategies to substantiate their findings. It is an important contribution and would be useful to a general readership in endocrinology, bone and fat metabolism.

We thank this reviewer for his/her warm words on the quality of our experiments.

1. The reviewer is correct that our previous studies (Zhu et al., PNAS and Ji *et al.*, *PNAS*), that used our polyclonal and mouse monoclonal FSH–blocking antibodies, respectively, did not reveal a significant reduction in serum FSH levels. Our current study using MS-Hu6 also did not show a significant drop in serum FSH. This new data with serum FSH is shown in Figure 1E, with an explanation in lines 107 to 111 of the revised manuscript.

2. We have now included serum total inhibin levels in Figure 1E with mention in the text in lines 107 to 109 (revised manuscript).

3. We have included data and images of cortical thickness (Ct.Th) in response to MS-Hu6 in Figure 1G and Supplementary Figure 1D of the revised manuscript and referred to these in the text (Results section, lines 119 to 123, revised manuscript).

Reviewer #2 (Recommendations for the authors):This study is a well-designed set of experiments as a comprehensive characterization of MS-Hu6 for its efficacy in mouse models of obesity and osteoporosis, acute safety in monkeys, a full evaluation of its pharmacokinetic, pharmacodynamic and biodistribution, and a compendium of its physicochemical properties for treatment of obesity and bone loss in people. The manuscript is also well written. The manuscript is ready for publication.

We are grateful to the reviewer for examining our manuscript. There is no request for change.